# SINS/Landmark Integrated Navigation Based on Landmark Attitude Determination

**DOI:** 10.3390/s19132917

**Published:** 2019-07-01

**Authors:** Shuqing Xu, Haiyin Zhou, Jiongqi Wang, Zhangming He, Dayi Wang

**Affiliations:** 1College of Liberal Arts and Sciences, National University of Defense Technology, Changsha 410073, China; 2Beijing Institute of Spacecraft System Engineering, China Academy of Space Technology, Beijing 100094, China

**Keywords:** SINS/Landmark integrated navigation, landmark attitude determination, all-day and all-weather navigation, image matching, navigation Kalman filter

## Abstract

Based on the situation that the traditional SINS (strapdown inertial navigation system)/CNS (celestial navigation system) integrated navigation system fails to realize all-day and all-weather navigation, this paper proposes a SINS/Landmark integrated navigation method based on landmark attitude determination to solve this problem. This integrated navigation system takes SINS as the basic scheme and uses landmark navigation to correct the error of SINS. The way of the attitude determination is to use the landmark information photographed by the landmark camera to complete feature matching. The principle of the landmark navigation and the process of attitude determination are discussed, and the feasibility of landmark attitude determination is analyzed, including the orthogonality of the attitude transform matrix, as well as the influences of the factors such as quantity and geometric position of landmarks. On this basis, the paper constructs the equations of the SINS/Landmark integrated navigation system, testifies the effectiveness of landmark attitude determination on the integrated navigation by Kalman filter, and improves the navigation precision of the system.

## 1. Introduction

In view of the advantages of strong autonomy, anti-interference and good concealment, SINS and CNS are widely used [1,2]. Inertial devices can provide the navigation data of attitude, velocity and position. Star sensors measure the information of astronomical object. This information can be combined and used to obtain more precise navigation results [3,4,5,6]. However, in real application, CNS is unable to work continuously throughout the day. In addition, it can be affected by bad weather, thus SINS/CNS integrated navigation cannot realize a continual navigation for a long time [7].

Recently, as a new navigation technology, landmark navigation has received wide attention globally. As can be seen, landmark navigation is technically different from CNS. It is based on the visualization of the landmarks, which can solve the drawbacks of star sensors and navigate the aircraft by using the geographical features of the planet’s surface [8,9]. To compensate the application limitation, this paper combines SINS with landmark navigation to construct a pattern with high precision to meet navigation requirements.

Currently, some studies have been conducted on the theory and application of landmark navigation. For the single landmark navigation, Costello and Castro [10] extracted landmark characteristics from 2D sensor images, and combined them with the landmark database to estimate the latitude, longitude and attitude of the vehicles on the ground or targets at space. Cesetti et al. [11] proposes a vision-based method of applying UAV (unmanned aerial vehicle) for landing. The basic principle of the guiding method is to define the target area by satellite images or antenna with high resolution . The key of the navigation strategy is to locate proper natural landmarks by feature matching algorithm.

For the applications of landmark navigation in practice, He et al. [12] proposed a method to obtain the information of the space target emitter by landmarks. The purpose is to obtain the azimuth angle of the target, and then calculate the attitude by planarization algorithm and linearization technique. In Khuller et al. [13], landmark navigation is applied in the field of robot industry. Robots can realize the position determination through the landmarks based on the direction information provided by visual inspection. It proposes the minimum set of landmarks for the robot position determination, and the quantity of this minimum set of landmarks is called “measurement dimension”.

However, landmark navigation is also limited by the weather and environment conditions, thus it should be integrated with other navigation methods to realize a more precious navigation. In the integration algorithm between the landmark navigation and other autonomous navigation methods, Kim and Hwang [14] raised the integration of SINS and landmark navigation. Its navigation mode is roughly the same as the one in this paper. It is designed to solve the problem of visual navigation under poor sight. When the environment makes it hard to observe landmarks, the landmark visual navigation is forced to interrupt; the inertial devices continue working and providing position, velocity and attitude information. This method can make up the shortfalls of single landmark navigation and realize a continuous navigation.

When the landmark navigation is combined with other non-autonomous navigations, Babel [15] proposed a navigation algorithm to determine the shortest route of UAV(unmanned aerial vehicle). The landmark navigation is used to compensate the navigation interruption of the GPS. Meanwhile, the method regularly updates the landmark base to realize long-term landmark navigation.

The above schemes all use landmarks for position determination, and the precision of their navigation results can be improved greatly. Under the practical research background, both the aircraft and the astronomical object are relatively rotating to the inertial system, thus it is hard to identify the real attitude of the aircraft to the fixed coordinate system of astronomical object. To achieve the best navigation result, the landmark information should be fully used to accomplish the attitude determination. Therefore, the key of the landmark navigation is to determine the attitude based on landmark characteristics.

In view of the autonomous navigation around the Earth, this paper proposes a SINS/Landmark integrated navigation method with high precision based on landmark attitude determination to meet the demands of the all-day and all-weather autonomous navigation. During the landmark navigation, landmark camera photographs the landmarks at its sampling frequency to capture the landmark characteristics. Then, the characteristics are recognized and matched with the landmark database to obtain the position and attitude information of the aircraft. Position and attitude information by the landmark navigation is combined with the measurement by inertial devices to get more precise navigation results. This paper also introduces the landmark navigation’s principle and attitude determination process, constructs the equations for SINS/Landmark integrated navigation system, compares the navigation results with those have non-attitude determination in the simulations, and testifies the effectiveness of landmark attitude determination on integrated navigation.

The main contributions of this paper include the following. (1) Combine landmark navigation with SINS: This paper proposes SINS/Landmark integrated navigation mode, which can continue navigating and realizing an all-day and all-weather navigation when the SINS/CNS mode is not working. (2) This method uses the landmark features obtained by the landmark camera to accomplish attitude determination, and combines the attitude information with the measurement of SINS to realize an integrated navigation with high precision. (3) This method applies Kalman filter to the linear navigation system, and illustrates the influence of the attitude determination on the navigation accuracy by comparison in the simulations.

The remainder of the paper is structured as follows. Section 2 introduces the principle of landmark navigation, including the acquisition and matching of landmark information. Section 3 introduces the process of determining the attitude by using the landmarks. In this process, the computability of the rotation matrix is verified, and the feasibility of the landmark attitude determination is theoretically explained. Section 4 introduces the SINS/Landmark integrated navigation and gives the system equations. In Section 5, the effectiveness of the SINS/Landmark integrated navigation with landmark attitude determination is verified by experimental simulation. Section 6 is the conclusion.

## 2. Principles of Landmark Navigation

### 2.1. Acquisition of Landmark Information

The landmark information can be obtained from the landmark camera, and the operation process is mainly divided into two parts: the shooting process of the landmark camera and the matching process of the landmark feature database.

It is known that the imaging principle of the camera is “small hole imaging”, and the landmark position vector obtained by the landmark information is actually the mapping point of the camera imaging focus, which is similar to the star sensor. However, in practical applications, the feature points of the mapping area may not be obvious and easy to extract. Therefore, the camera chooses the landmarks with clear features as the selection set, which is used to replace the virtual landmark of the mapping point. As for the equations of the position and attitude determination, there is no difference between the two selections.

As shown in Figure 1, when the aircraft (navigation target) orbits around the Earth, the landmark camera first captures the landmark images, and the images are transmitted to the information processing module. Then, the features are compared and analyzed, including the position information of the landmarks on the Earth, and then follows up. To keep the navigation going, the duration of the matching process should be less than the shooting time of the camera.

As shown in Figure 1, there are the processes of shooting, landmark capture and landmark information acquisition, respectively.

(**a**) Firstly, shoot the landmarks by the landmark camera. It should be noted that only one photo can be taken at a time depending on the shooting frequency of the camera.

(**b**) Secondly, the landmark camera obtains the images of the landmarks. At this time, multiple landmarks and landmark feature points are obtained according to the photos, and these feature points participate in the next step for matching.

(**c**) Finally, the landmark feature points in the photo are extracted and then matched with the feature points database. When the matching is completed, the aircraft’s space model is established for navigation. The space model is shown in the third picture of Figure 1. Since the vectors in the figure are all referenced to the Earth-centered-Earth-fixed system, the superscript of the vectors *e*, $\rho_{1}^{e}$ and $\rho_{2}^{e}$ are the position coordinates of the landmarks; $\rho_{0}^{e}$ is the vector from Landmark 1 to Landmark 2; $r^{e}$ is the position coordinates of the aircraft; and $r_{1}^{e}$ and $r_{2}^{e}$ are, respectively, the vectors from Landmark 1 and Landmark 2 to the aircraft, i.e., the relative position model of the aircraft and landmarks at space.

When the aircraft is not parallel to the landmark flat, the captured image of the landmark is prone to rotation and deformation as shown IN Figure 2:

When the landmark camera is not parallel to the landmark flat, the segment ab of the landmark is imaged as *g* on the imaging area of the camera. According to the principle that the proportion of the similar triangle is constant, the imaging of the segment ab in the virtual flat should be *h*, and then the image compresses in the O direction, the compression proportion κ is as Equation (Equation 1):(1)$$\kappa =\frac{g}{h}$$

When the landmark camera is fixed on the aircraft, the transformation matrix between the landmark camera coordinate system and the aircraft coordinate system is a constant matrix. When the landmark camera changes direction with the flight of the aircraft, the transformation matrix will also be changed. In practice, the transformation matrix needs to be selected and calculated according to the situation.

### 2.2. Matching Process of the Landmark Images

The image rotation and deformation should be considered when matching the landmarks, which will take more time. In practice, this process can be operated by using sparse representation method or feature point detection method.

The sparse representation method is to express few original signals to the total by using a linear combination of these original signals [16,17,18], but it takes a long time, which cannot meet the real-time requirement. The feature point detection method is widely used in the visual-based celestial landing navigation. SIFT (Scale-Invariant Feature Transform) is one of the popular methods with preferable application effect [19,20], which realizes the identification and matching of landmark images by key point detection, description, matching and elimination of mismatch points. This algorithm has good discriminability, strong scalability and better robustness in the case of target rotation and deformation, and it can extract multiple features of the target.

For the matching process of landmarks, the operational process of the SIFT is shown in Figure 3:

The scale space theory is used to search the feature points [21], and the Gaussian pyramid [22] and the Gaussian difference pyramid need to be established to compare with the pixels. The gradient of neighborhood feature points needs to be calculated and the main direction leads to the peak. The KD tree [23] is used for the matching process and the threshold should be set by the circumstances.

## 3. Position and Attitude Determination of the Landmark Navigation

### 3.1. Attitude Determination of the Landmark Navigation

#### 3.1.1. Calculation for Attitude Angle

The coordinate systems used in this paper are as follows: *i* is the inertial coordinate system; *e* is the Earth-centered-Earth-fixed system; *b* is the aircraft body coordinate system; and li is the launching inertial system.

Assume that the position of the aircraft is $r^{e}=\left(x^{e},y^{e},z^{e}\right)$ in the Earth-centered-Earth-fixed system, the position of the observed landmark is $\rho_{i}^{e}=\left(a_{i}^{e},b_{i}^{e},c_{i}^{e}\right)i=1,\cdots ,\tau$, representing that τ landmark can be observed. Assume that the distance between the aircraft and each landmark point is $r_{i}$, and $\left(\alpha_{i},\beta_{i},\eta_{i}\right)$ is the angle between the axes of the aircraft body coordinate system and the vector from the aircraft to the landmark. $C_{e}^{b}$ is the transformation matrix from the Earth-centered-Earth-fixed system to the aircraft body coordinate system, then:(2)$$C_{e}^{b}{\left[\begin{array}{ccc} \frac{x^{e}-a_{i}^{e}}{r_{i}} & \frac{y^{e}-b_{i}^{e}}{r_{i}} & \frac{z^{e}-c_{i}^{e}}{r_{i}} \end{array}\right]}^{T}=\left[\begin{array}{c} \cos\alpha_{i}\\ \cos\beta_{i}\\ \cos\eta_{i} \end{array}\right]$$

Let
(3)$${\left[\begin{array}{ccc} \frac{x^{e}-a_{i}^{e}}{r_{i}} & \frac{y^{e}-b_{i}^{e}}{r_{i}} & \frac{z^{e}-c_{i}^{e}}{r_{i}} \end{array}\right]}^{T}=X_{i},\left[\begin{array}{c} \cos\alpha_{i}\\ \cos\beta_{i}\\ \cos\eta_{i} \end{array}\right]=Z_{i}$$

Because of the measurement error, according to Equation (Equation 2), it can be written as Equation (Equation 4):(4)$$Z=C_{e}^{b}X+\varepsilon$$
where $X={\left[X_{1},\cdots X_{\tau }\right]}^{T}$, $Z={\left[Z_{1},\cdots Z_{\tau }\right]}^{T}$, $\varepsilon ={\left[\varepsilon_{1},\cdots \varepsilon_{\tau }\right]}^{T}∼N\left(0,\sigma^{2}\right)$. If τ≥3, the transformation matrix can be calculated by the least squares (LS) method as Equation (Equation 5):(5)$$C_{e}^{b}=ZX^{T}{\left(XX^{T}\right)}^{-1}$$

The Launch point’s coordinate in the geodetic coordinate system is $\left(B,L,H\right)$, and the angle of incidence is *A*, then the transformation matrix between the launch inertial system and the geocentric inertial system can be calculated as Equation (Equation 6):(6)$$C_{li}^{i}=\left[\begin{array}{ccc} \sin L & \cos L & 0\\ -\cos L & \sin L & 0\\ 0 & 0 & 1 \end{array}\right]\left[\begin{array}{ccc} 1 & 0 & 0\\ 0 & \cos B & -\sin B\\ 0 & \sin B & \cos B \end{array}\right]\left[\begin{array}{ccc} -\sin A & 0 & -\cos A\\ 0 & 1 & 0\\ \cos A & 0 & -\sin A \end{array}\right]$$

The transformation matrix between the geocentric inertial system and the Earth-centered-Earth-fixed system is
(7)$$C_{i}^{e}=\left[\begin{array}{ccc} \cos\left(\omega T\right) & \sin\left(\omega T\right) & 0\\ -\sin\left(\omega T\right) & \cos\left(\omega T\right) & 0\\ 0 & 0 & 1 \end{array}\right]$$
where ω is the angular velocity rate of the Earth’s rotation, and *T* is the rotation time of the Earth. At this point, the transformation matrix between the launch inertial system and the aircraft body coordinate system can be obtained as Equation (Equation 8):(8)$$C_{li}^{b}=C_{e}^{b}C_{i}^{e}C_{li}^{i}$$

Then, the pitch angle, yaw angle and roll angle of the aircraft can be obtained, separately.
(9)φ=arctanCbli2,1Cbli2,1Cbli1,1Cbli1,1ϕ=−arcsinCbli3,1γ=arctanCbli3,2Cbli3,2Cbli3,3Cbli3,3

$C_{b}^{li}$ is the transposition of $C_{li}^{b}$, representing the transformation matrix from the aircraft body coordinate system to the launch inertial system.

#### 3.1.2. Computability of the Transformation Matrix

It is necessary to prove the computability of the transformation matrix, but its orthogonal should be proved first. In theory, the transformation matrix is orthogonal obviously, representing the rotation relationship between the two coordinate systems. However, in this paper, the transformation matrix $C_{e}^{b}$ is obtained by the distance scalar $r_{i}$ of the aircraft and each landmark point, as well as the angle $\left(\alpha_{i},\beta_{i},\eta_{i}\right)$ between the coordinate axes of the aircraft body coordinate system, and the vector from the aircraft to the landmark, thus it is significant to prove its orthogonality.

The computability of the transformation matrix between the launch inertial system and the aircraft body coordinate system is equivalent to that of the transformation matrix $C_{e}^{b}$ from the Earth-centered-Earth-fixed system to the aircraft body coordinate system. That is, Equation (Equation 5) can be calculated.

According to Equation (Equation 5), the matrix X must be full row rank, that is, the vector from the aircraft to the landmark point can be expanded into the entire space.

Proof for the orthogonality of matrix $C_{e}^{b}$:

(1) Invariance of the mold length: Let $r_{i}$ be the mold length of ${\left[\begin{array}{ccc} x^{e}-a_{i}^{e} & y^{e}-b_{i}^{e} & z^{e}-c_{i}^{e} \end{array}\right]}^{T}$, and it can be known that the mold length of the column vectors in matrix X are all 1, and $\left(\alpha_{i},\beta_{i},\eta_{i}\right)$ is the angle between the three axes of the aircraft body coordinate system and the vector from the aircraft to the landmark. The column vector mold length of Z is also 1.

(2) Invariability of the angles: For any two landmarks (set to be the *i*th and the *j*th), the vectors from aircraft to the two landmarks can be set as $r_{i}^{e}$ and $r_{j}^{e}$ of the Earth-centered-Earth-fixed system, respectively. $r_{i}^{b}$ and $r_{j}^{b}$ are, respectively, the two landmarks of the aircraft body coordinate system, satisfying
(10)$$\left\{\begin{array}{c} r_{i}^{b}=C_{e}^{b}r_{i}^{e}\\ r_{j}^{b}=C_{e}^{b}r_{j}^{e} \end{array}\right.$$

It is known that the same angle does not change in size at different coordinates, and consider that
(11)$$\left\{\begin{array}{c} r_{i}^{e}r_{j}^{e}=\left|r_{i}^{e}\right|\left|r_{j}^{e}\right|\cos\langle r_{i}^{e},r_{j}^{e}\rangle =\cos\langle r_{i}^{e},r_{j}^{e}\rangle \\ r_{i}^{b}r_{j}^{b}=\left|r_{i}^{b}\right|\left|r_{j}^{b}\right|\cos\langle r_{i}^{b},r_{j}^{b}\rangle =\cos\langle r_{i}^{b},r_{j}^{b}\rangle \end{array}\right.$$

Then,
(12)$$r_{i}^{e}r_{j}^{e}=r_{i}^{b}r_{j}^{b}$$

For convenience, taking three sets of linear independent column vectors in two coordinate systems, Equation (Equation 4) can be rewritten as Equation (Equation 13):(13)$$C_{e}^{b}\tilde{X}=\tilde{Z}$$
where $\tilde{X}={\left[r_{k_{1}}^{e},r_{k_{2}}^{e},r_{k_{3}}^{e}\right]}^{T}$, $\tilde{Z}={\left[r_{k_{1}}^{b},r_{k_{2}}^{b},r_{k_{3}}^{b}\right]}^{T}$, $k_{1},k_{2},k_{3}$ are three randomly selected landmarks. According to Equation (Equation 12),
(14)X˜TX˜=rkie·rkjei=1,2,3j=1,2,3=rkib·rkjbi=1,2,3j=1,2,3=Z˜TZ˜

Therefore,
(15)$$\tilde{Z}{\tilde{X}}^{-1}={\left({\tilde{Z}}^{-1}\right)}^{T}{\tilde{X}}^{T}$$

Calculating the inverse matrix of $C_{e}^{b}$, Equation (Equation 16) can be obtained after transposition:(16)$${\left[{\left(C_{e}^{b}\right)}^{-1}\right]}^{T}={\left({\tilde{Z}}^{-1}\right)}^{T}{\tilde{X}}^{T}$$

Considering Equation (Equation 13),
(17)$$C_{e}^{b}={\left({\tilde{Z}}^{-1}\right)}^{T}{\tilde{X}}^{T}$$

According to Equations (Equation 16) and (Equation 17),
(18)$${\left(C_{e}^{b}\right)}^{T}={\left(C_{e}^{b}\right)}^{-1}$$

Therefore, the matrix $C_{e}^{b}$ is orthogonal. Moreover, it is computable.

#### 3.1.3. The Relationship between the Number, Relevance and Accuracy of Landmarks

The selection of landmarks will affect the accuracy of navigation. The influential factors mainly contain the number of landmarks and the relevance of landmark locations (whether they are collinear or not), etc. The two aspects are discussed and the analysis results are given as follows.

From Equation (Equation 4), the estimation error of the transformation matrix $C_{e}^{b}$ is as Equation (Equation 19):(19)$$Cov\left(C_{e}^{b}\right)={\left(XX^{T}\right)}^{-1}\sigma^{2}$$


**The relationship between the number of landmarks and the accuracy:**


According to Gorbenko and Popov [24], for the set of landmarks, there is usually a minimum set of them. Meanwhile, the completeness can be satisfied. In the process of the position determination, the purpose is to obtain the coordinate of the aircraft. In general, a three-dimensional coordinate has three unknowns elements, and this means the number of observable landmarks should be three or more. The specific proof and analysis can refer to Section 3.2.

However, for the process of attitude determination, since the solution of the transformation matrix is calculated by the LS method using multiple samples, the relationship between the selection of the landmarks and the accuracy of the attitude needs to be determined according to Equation (Equation 19).

The following part is the analysis of the number of selected landmarks in the attitude determination.

Assuming that a new landmark is added on the basis of several original landmarks, Equation (Equation 19) can be rewritten as
(20)$$\begin{array}{c} Cov\left(C_{e}^{b}\right)={\left(\left[\begin{array}{cc} X & X_{\tau +1} \end{array}\right]\left[\begin{array}{c} X^{T}\\ X_{\tau +1}^{T} \end{array}\right]\right)}^{-1}\sigma^{2}\\ \;\;\;\;\;\;\;\;\;\;\;\;\;\;\;\;\;={\left(XX^{T}+X_{\tau +1}X_{\tau +1}^{T}\right)}^{-1}\sigma^{2} \end{array}$$

Since the eigenvalues of the matrix ${\left(XX^{T}+X_{\tau +1}X_{\tau +1}^{T}\right)}^{-1}$ are smaller than the eigenvalues of the matrix ${\left(XX^{T}\right)}^{-1}$, within the calculation amount, a larger number of landmarks means higher accuracy. Since the landmark camera can only get one picture at a time according to its shooting frequency, and the number of landmarks in the picture is limited, it can be concluded that the more landmarks there are, the higher is the accuracy of the attitude that can be achieved.


**The relationship between the landmark correlation and the accuracy:**


If there is a strong correlation between the landmarks, even the landmarks are selected as points near a line, the minimum eigenvalue of the matrix $XX^{T}$ tends to zero. Make an eigenvalue decomposition (ED) of the matrix:(21)$${\left(XX^{T}\right)}^{-1}=P\Lambda^{-1}P^{-1}$$

Then, the mean square error (MSE) of the transformation matrix $C_{e}^{b}$ is as Equation (Equation 22):(22)$$\begin{array}{c} MSE\left(C_{e}^{b}\right)=\sigma^{2}\mathrm{tr}{\left(XX^{T}\right)}^{-1}=\sigma^{2}\mathrm{tr}\left(P\Lambda^{-1}P^{-1}\right)\\ \;\;\;\;\;\;\;\;\;\;\;\;\;\;\;\;\;\;\;\;=\sigma^{2}\mathrm{tr}\left(\Lambda^{-1}PP^{-1}\right)=\sigma^{2}\mathrm{tr}\left(\Lambda^{-1}\right)\\ \;\;\;\;\;\;\;\;\;\;\;\;\;\;\;\;\;\;\;\;=\sigma^{2}\sum_{i=1}^{3}\lambda_{i}^{-1} \end{array}$$

Since the landmarks are considered in three-dimensional space, the eigenvalue matrix is $\Lambda =\left[\begin{array}{ccc} \lambda_{1} & & \\ & \lambda_{2} & \\ & & \lambda_{3} \end{array}\right]$, and P is the eigenvector matrix.

Since $\lambda_{3}$ approaches 0, when the matrix is inverted, $MSE\left(C_{e}^{b}\right)$ tends to infinity. At this time, the calculation error is particularly large, which affects the calculation accuracy.

The simulation data were used to give an explanation.

Assuming that the measurement angle error obeys the normal distribution with a zero mean value and a standard deviation of 30”, the errors can be calculated under the different number of landmarks and correlation. In each case, Monte Carlo simulation was performed for 100 times to obtain the mean error. The accuracy results are shown in Table 1:

The following can be seen in Table 1:

(a) When there is a strong correlation between the landmarks, the angle error becomes larger, especially the pitch angle error.

(b) Without considering the correlation of the landmarks, more landmarks make the angle error smaller. That is, the more accurate is the transformation matrix, the higher is the accuracy of the attitude determination. This shows that, in the actual navigation process, more landmarks lead to higher accuracy.

(c) In the case that a gradual increase emerges in the number of landmarks, whether there is a correlation relationship between landmarks has less and less influence on the accuracy of the transformation matrix. This shows that, when the number increases, the collinear problem inevitably appears. How to balance the number and the location of the landmarks is the key to improving the navigation accuracy.

### 3.2. Position Determination of Landmark Navigation

It is shown in Section 3.1 that the position of the *i*th navigation landmark in the Earth-centered-Earth-fixed system is $\rho_{i}^{e}=\left(a_{i}^{e},b_{i}^{e},c_{i}^{e}\right),\;i=1,\cdots ,\tau$, the vector of the aircraft under the Earth-centered-Earth-fixed system is $r^{e}=\left(x^{e},y^{e},z^{e}\right)$, and the relative position of the aircraft to the landmark is $r_{i}^{e}$. Let the transformation matrix of the aircraft body coordinate system to the Earth-centered-Earth-fixed system be $C_{b}^{e}={\left(C_{e}^{b}\right)}^{T}$, and $C_{e}^{b}$ can be obtained from Section 3.1.1. Then,
(23)$$r_{i}^{e}={\left(\begin{array}{ccc} x^{e}-a_{i}^{e} & y^{e}-b_{i}^{e} & z^{e}-c_{i}^{e} \end{array}\right)}^{T}$$

Suppose that $\rho_{i}^{b}$ is the position vector of the *i*th landmark in the aircraft body coordinate system, which can be obtained by the landmark camera. Then, it can be obtained as Equation (Equation 24):(24)$$\begin{array}{c} r_{i}^{e}=C_{b}^{e}\rho_{i}^{b}\\ \Rightarrow \left[\begin{array}{c} x^{e}-a_{i}^{e}\\ y^{e}-b_{i}^{e}\\ z^{e}-c_{i}^{e} \end{array}\right]=C_{b}^{e}\left[\begin{array}{c} a_{i}^{e}\\ b_{i}^{e}\\ c_{i}^{e} \end{array}\right] \end{array}$$

Suppose that $x^{e},\;y^{e},\;z^{e}$ are three unknown vectors, and the coordinate of the aircraft in the Earth-centered-Earth-fixed system can be obtained when there are three or more landmarks can be observed, that is τ≥3.

## 4. SINS/Landmark Integrated Navigation Model

The operation process of SINS/Landmark integrated navigation is based on SINS, and the landmark information is used to correct the error of the SINS. The measurement equation is established according to the position and attitude information obtained by SINS and landmark navigation. The specific navigation process is shown in Figure 4.

### 4.1. State Equation of Integrated Navigation

Combined with the SINS mathematical platform angle error, velocity error, position error, gyroscope and the accelerometer error models, the state equation of the integrated navigation system can be obtained as Equation (Equation 25):(25)$$\dot{X}\left(t\right)=F\left(t\right)X\left(t\right)+G\left(t\right)W\left(t\right)$$

Take the state parameter of the system as 15 dimensions, and record it as:(26)$$X\left(t\right)={\left[\phi_{x},\phi_{y},\phi_{z},\delta v_{x},\delta v_{y},\delta v_{z},\delta x,\delta y,\delta z,\varepsilon_{x},\varepsilon_{y},\varepsilon_{z},\nabla_{x},\nabla_{y},\nabla_{z}\right]}^{T}$$
where ${\left[\begin{array}{ccc} \phi_{x} & \phi_{y} & \phi_{z} \end{array}\right]}^{T}$ denotes the three mathematical platform angles error on three axes; ${\left[\begin{array}{ccc} \delta v_{x} & \delta v_{y} & \delta v_{z} \end{array}\right]}^{T}$ denotes the velocity error; ${\left[\begin{array}{ccc} \delta x & \delta y & \delta z \end{array}\right]}^{T}$ denotes the position error; $\left[\begin{array}{ccc} \varepsilon_{x} & \varepsilon_{y} & \varepsilon_{z} \end{array}\right]$ denotes the three random constant drift of gyroscope; ${\left[\begin{array}{ccc} \nabla_{x} & \nabla_{y} & \nabla_{z} \end{array}\right]}^{T}$ denotes the three random constant bias of accelerometer; $F\left(t\right)$ denotes the process input matrix; and $G\left(t\right)$ denotes the process noise matrix.
(27)$$F\left(t\right)={\left[\begin{array}{ccccc} 0_{3\times 3} & 0_{3\times 3} & 0_{3\times 3} & C_{b}^{li} & 0_{3\times 3}\\ F_{b} & 0_{3\times 3} & F_{a} & 0_{3\times 3} & C_{b}^{li}\\ 0_{3\times 3} & I_{3\times 3} & 0_{3\times 3} & 0_{3\times 3} & 0_{3\times 3}\\ 0_{3\times 3} & 0_{3\times 3} & 0_{3\times 3} & 0_{3\times 3} & 0_{3\times 3}\\ 0_{3\times 3} & 0_{3\times 3} & 0_{3\times 3} & 0_{3\times 3} & 0_{3\times 3} \end{array}\right]}_{15\times 15}$$
(28)$$G\left(t\right)={\left[\begin{array}{cc} C_{b}^{li} & 0_{3\times 3}\\ 0_{3\times 3} & C_{b}^{li}\\ 0_{3\times 3} & 0_{3\times 3}\\ 0_{3\times 3} & 0_{3\times 3}\\ 0_{3\times 3} & 0_{3\times 3} \end{array}\right]}_{15\times 6}$$
where $C_{b}^{li}$ is the transformation matrix from the aircraft body coordinate system to the launch inertial system, and
(29)$$C_{b}^{li}=\left[\begin{array}{ccc} \cos\theta \cos\varphi & \cos\theta \sin\varphi \sin\gamma -\sin\theta \cos\gamma & \cos\theta \sin\varphi \cos\gamma +\sin\theta \sin\gamma \\ \sin\theta \cos\varphi & \sin\theta \sin\varphi \sin\gamma +\cos\theta \cos\gamma & \sin\theta \sin\varphi \cos\gamma -\cos\theta \sin\gamma \\ -\sin\varphi & \sin\gamma \cos\varphi & \cos\gamma \cos\varphi \end{array}\right]$$
where φ,θ,γ are the yaw angle, pitch angle and roll angle of the aircraft measured by the gyroscope, respectively.

$F_{a}=\left[\begin{array}{ccc} f_{14} & f_{15} & f_{16}\\ f_{24} & f_{25} & f_{26}\\ f_{34} & f_{35} & f_{36} \end{array}\right]$. The parameters $f_{14}$–$f_{36}$ are the derivative of gravitational acceleration on position coordinates, and they vary with the position of the aircraft.

$F_{b}=\left[\begin{array}{ccc} 0 & -a_{z} & a_{y}\\ a_{z} & 0 & -a_{x}\\ -a_{y} & a_{x} & 0 \end{array}\right]$. $a_{x},a_{y},a_{z}$ are the components of the apparent acceleration along the three axes according to the accelerometer. The process noise of the navigation system is as Equation (Equation 30):(30)$$W\left(t\right)={\left[\begin{array}{cccccc} w_{\varepsilon_{x}} & w_{\varepsilon_{y}} & w_{\varepsilon_{z}} & w_{\nabla_{x}} & w_{\nabla_{y}} & w_{\nabla_{z}} \end{array}\right]}^{T}$$
where ${\left[\begin{array}{ccc} w_{\varepsilon_{x}} & w_{\varepsilon_{y}} & w_{\varepsilon_{z}} \end{array}\right]}^{T}$ is the random noise of gyro; and ${\left[\begin{array}{ccc} w_{\nabla_{x}} & w_{\nabla_{y}} & w_{\nabla_{z}} \end{array}\right]}^{T}$ is the random noise of accelerometer. The noise covariance matrix of $W\left(t\right)$ is
(31)$$Q\left(t\right)=diag\left(\left[\begin{array}{cccccc} \sigma_{\varepsilon_{x}}^{2} & \sigma_{\varepsilon_{y}}^{2} & \sigma_{\varepsilon_{z}}^{2} & \sigma_{\nabla_{x}}^{2} & \sigma_{\nabla_{y}}^{2} & \sigma_{\nabla_{z}}^{2} \end{array}\right]\right)$$

### 4.2. Measurement Equation of Integrated Navigation

According to the landmark navigation process, the output is the attitude and position information of the aircraft, which subtracts the attitude and position information measured from the SINS to obtain the measurement equation. Therefore, the observation of the measurement equation is the transformed platform angle error (the navigation frame misalignment angle) $\phi_{x},\;\phi_{y},\;\phi_{z}$ and position error δx,δy,δz.


**Solution process of the mathematical platform misalignment angle $\phi_{x},\;\phi_{y},\;\phi_{z}$:**


In the SINS/Landmark integrated navigation process, SINS obtains the pitch angle $\theta_{0}$, yaw angle $\varphi_{0}$ and roll angle $\gamma_{0}$ of the aircraft through the strap down solution, and the landmark camera obtains the pitch angle θ, yaw angle φ and roll angle γ, subtracting the two sets of angles to obtain the three-axis attitude error as Equation (Equation 32):(32)$$\Delta a=\left[\begin{array}{c} a_{\theta }\\ a_{\varphi }\\ a_{\gamma } \end{array}\right]=\left[\begin{array}{c} \theta -\theta_{0}\\ \varphi -\varphi_{0}\\ \gamma -\gamma_{0} \end{array}\right]$$

Since the composition of the attitude error equation of SINS is the platform angle error (the navigation frame misalignment angle), it is necessary to convert the attitude error angle of Equation (Equation 32) into the platform angle error to establish the measurement equation. The conversion relationship is as Equation (Equation 33):(33)$$\Delta a^{\prime }=M_{1}\Delta a$$
where $M_{1}$ is the attitude angle error transformation matrix, and $\Delta a^{\prime }={\left[\begin{array}{ccc} \phi_{x} & \phi_{y} & \phi_{z} \end{array}\right]}^{T}$ is the platform angle error.


**Solution process of the position error δx,δy,δz:**


The position coordinate $r^{e}={\left[\begin{array}{ccc} x^{e} & y^{e} & z^{e} \end{array}\right]}^{T}$ of the aircraft under the Earth-centered-Earth-fixed system can be obtained by the position determination process of landmark navigation as in Section 3.2. The coordinate of the aircraft under the inertial system of the Earth’s core, i.e., $r^{i}={\left(\begin{array}{ccc} x^{i} & y^{i} & z^{i} \end{array}\right)}^{T}$, can be calculated by the inertial device. Then,
(34)$${\tilde{r}}^{i}=C_{e}^{i}r^{e}$$

The position error under the inertial system is as Equation (Equation 35):(35)$$\delta r^{i}=\left[\begin{array}{c} \delta x^{i}\\ \delta y^{i}\\ \delta z^{i} \end{array}\right]=r^{i}-{\tilde{r}}^{i}$$

Converting it to the mathematical platform coordinate system, the conversion relationship is as Equation (Equation 36):(36)$$\delta r=M_{2}\delta r^{i}$$
where $M_{2}$ is the position error conversion matrix, and $\delta r={\left[\begin{array}{ccc} \delta x & \delta y & \delta z \end{array}\right]}^{T}$ is the position error angle.

Therefore, the measurement equation is as Equation (Equation 37):(37)$$Z\left(t\right)=\left[\begin{array}{c} \phi_{x}\\ \phi_{y}\\ \phi_{z}\\ \delta x\\ \delta y\\ \delta z \end{array}\right]=\mathrm{HX}\left(t\right)+V\left(t\right)$$
where $H={\left[I_{3\times 3},0_{3\times 3},I_{3\times 3},0_{3\times 3}\right]}^{T}$, $V={\left[\delta \Delta_{x},\delta \Delta_{y},\delta \Delta_{z},\delta \xi_{x},\delta \xi_{y},\delta \xi_{z}\right]}^{T}$, ${\left[\delta \Delta_{x},\delta \Delta_{y},\delta \Delta_{z}\right]}^{T}$ denotes the difference between the attitude measurement noise of the landmark camera and the constant drift error of the gyroscope. ${\left[\delta \xi_{x},\delta \xi_{y},\delta \xi_{z}\right]}^{T}$ is the difference between position measurement noise of the landmark camera and the constant error of the accelerometer.

### 4.3. Integrated Navigation Filtering Algorithm

Consider the integrated navigation system according to Equations (Equation 25) and (Equation 37), discretize it and use Kalman filter to estimate it. The system equation can be transformed as Equation (Equation 38):(38)$$\left\{\begin{array}{c} X_{k+1}=\Phi_{k}X_{k}+\Gamma W_{k}\\ Z_{k}=HX_{k}+V_{k} \end{array}\right.$$

Assuming that *T* is the sampling interval, then
(39)$$\begin{array}{c} \Phi_{k}=I+F_{k-1}T+\frac{1}{2!}F_{k-1}^{2}T^{2}\\ \Gamma_{k-1}=T\left(I+\frac{1}{2!}F_{k-1}T+\frac{1}{3!}F_{k-1}^{2}T^{2}\right)G_{k} \end{array}$$

The Kalman filter algorithm is expressed as follows:

The one-step prediction equation:(40)$${\hat{X}}_{k+1|k}=\Phi_{k}X_{k}$$

The state estimation:(41)$${\hat{X}}_{k+1}={\hat{X}}_{k+1\left|k\right.}+K_{k}\left(Z_{k+1}-H{\hat{X}}_{k+1|k}\right)$$

The gain of the filter:(42)$$K_{k}=P_{k+1|k}H^{T}{\left(HP_{k+1|k}H^{T}+R_{k}\right)}^{-1}$$

The MSE of the one-step prediction:(43)$$P_{k+1|k}=\Phi_{k}P_{k}\Phi_{k}^{T}+\Gamma_{k}Q_{k+1}\Gamma_{k}^{T}$$

The MSE of the estimation:(44)$$P_{k+1}=\left(I-K_{k+1}H\right)P_{k+1|k}$$

The above are the basic equations of the discrete Kalman filter. When the initial value ${\hat{X}}_{0}$ and $P_{0}$ are given, combined with the measurement $Z_{k}$ of the *k* moment, the state estimation ${\hat{X}}_{k}$ at *k* moment can be obtained through iteration.

## 5. Simulation and Analysis

The Shandong Peninsula, the Liaodong Peninsula and the Yangtze River Delta were selected as the navigation landmarks. The landmark cameras were used to photograph the landmarks in the flight segment of the aircraft, and then the landmarks were compared to obtain the position and attitude of the aircraft. The matching process is shown in Figure 5, Figure 6 and Figure 7.

The trajectory of the spacecraft was drawn based on the captured information, as shown in Figure 8, Figure 9 and Figure 10.

The spacecraft was analyzed in the simulation, and the error parameters of SINS and landmark navigation were set as follows.

① Initial position errors of the three directions wer all 0 m. ② The initial speed errors were 0 m/s. ③ The initial attitude angle error was 0${}^{\prime \prime }$. ④ The gyro random constant drift was $0.1^{\circ }/h$. ⑤ The gyroscope random noise drift satisfied a normal distribution with a mean of 0 and a standard deviation of 0.5${}^{\prime \prime }$. ⑥ The accelerometer random constant bias was $10^{-4}g$, where *g* is the Earth gravity constant. ⑦ The accelerometer random noise offset satisfied a normal distribution with a mean of 0 and a standard deviation of $5\times 10^{-5}g$. ⑧ The standard deviation of the landmark navigation output position error was 50 m. ⑨ The standard deviation of the landmark navigation output attitude error was 3${}^{\prime \prime }$.

The sampling interval of SINS was 0.01 s, the landmark navigation was 0.1 s, the combined interval was 0.1 s, the simulation time was 1110 s, and the Monte Carlo method was used for simulation. Due to the requirements of the launch inertial system, initial latitude and longitude were $116.34^{\circ }$ and $39.98^{\circ }$, respectively, and the height was 0 m. The platform angle error, gyro drift, position error and speed error were corrected by the feedback during the navigation process.

To verify the importance of the landmark attitude determination in the integrated navigation, two methods (with attitude determine and without attitude determine) were simulated, and the trajectories of the aircraft was obtained, as shown in Figure 11.

It can be seen in Figure 11 that, during the whole navigation process, the orbit determination of the aircraft with attitude determination is almost the same as the real orbit, while the trajectory without attitude determination is far from the real orbit, which indicates that the attitude determination is critical. The comparison of the position as well as the velocity determination in three directions are given, respectively.

In Figure 12 , we can find that, in the X direction, when there is no attitude determination, the error of the position gradually becomes larger before 900 s, and there is a convergence trend after 900 s: the velocity error becomes larger with time. With the attitude determination, the position and velocity errors can be kept near 0. In Figure 13, for Y direction, without the attitude determination, the position and velocity errors diverge over time. With the attitude determination, the position error is very small, almost at 0 level and the speed figure has a slight error, but it also has a better improvement. In Figure 14, for Z direction, without the attitude determination, the position error diverges with time, the velocity error is divergent before 160 s, and then gradually converges after 160 s. When with the attitude determination, the changes of the position and velocity errors are similar to the Y direction.

Through the comparison of the three directions, it can be seen that the position errors gradually increase with time when the landmark only provides the positioning information, and large offsets appear in three directions, leading the gradually decreasing of the navigation accuracy. When the landmark completes the attitude determination, the navigation result can remain stable for a long time. The velocity determination is similar to the position determination.

Figure 12, Figure 13 and Figure 14 express the qualitative analysis of the accuracy on whether determine the attitude in the three directions, while Table 2 and Table 3 express the quantitative accuracy analysis of it.

It can be seen in Table 2 and Table 3 that it is quite different whether there is the attitude determination for navigation, and the attitude determination has a large impact on the navigation accuracy. When the attitude determination process is completed, the position and the velocity of the integrated navigation maintain at a high precision, and the navigation result approximates the true.

## 6. Conclusions

In this paper, faced with the limitation of the traditional SINS/CNS integrated navigation that the system is difficult to navigate all day and all weather, a SINS/Landmark integrated navigation method based on landmark attitude determination is proposed. Through the theoretical analysis of the landmark navigation principle and the simulation of the SINS/Landmark integrated navigation system, the landmark information is necessary to determine the attitude. When the landmark navigation only extracts the position information, the accuracy of the integrated navigation will maintain a large range of fluctuations. After the attitude determination, the accuracy of the integrated navigation can reach a high level and does not decrease with time. This shows that the SINS/Landmark integrated navigation method based on the landmark attitude determination is indeed effective, which can improve the accuracy of the navigation to some extent.

## Figures and Tables

**Figure 1 sensors-19-02917-f001:**
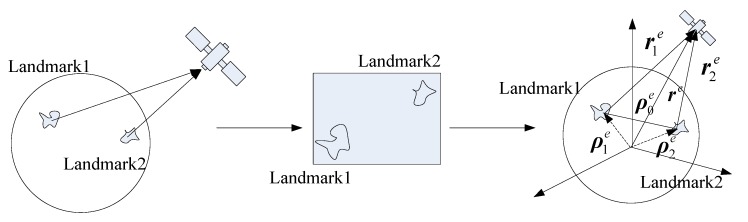
Acquisition process of landmarks.

**Figure 2 sensors-19-02917-f002:**
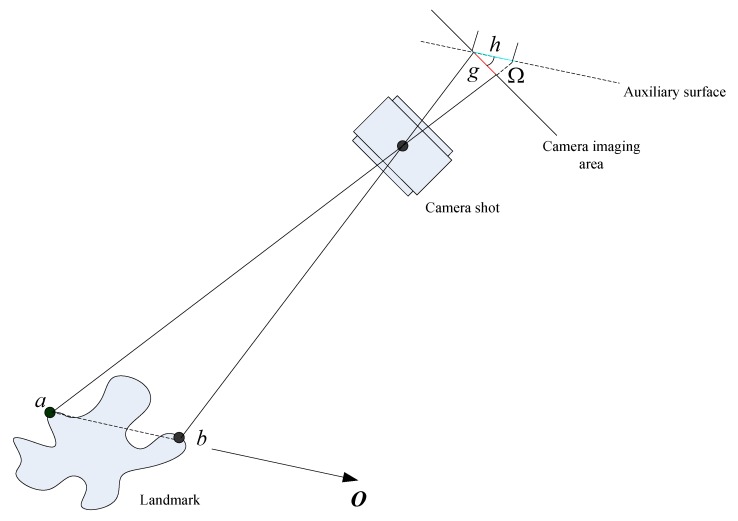
The principle of image compression during the landmark matching.

**Figure 3 sensors-19-02917-f003:**
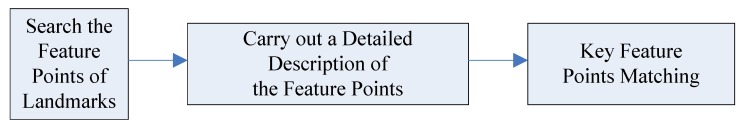
Flow of SIFT.

**Figure 4 sensors-19-02917-f004:**
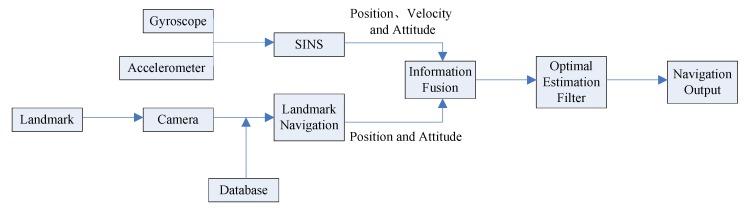
Process of the SINS/Landmark integrated navigation.

**Figure 5 sensors-19-02917-f005:**
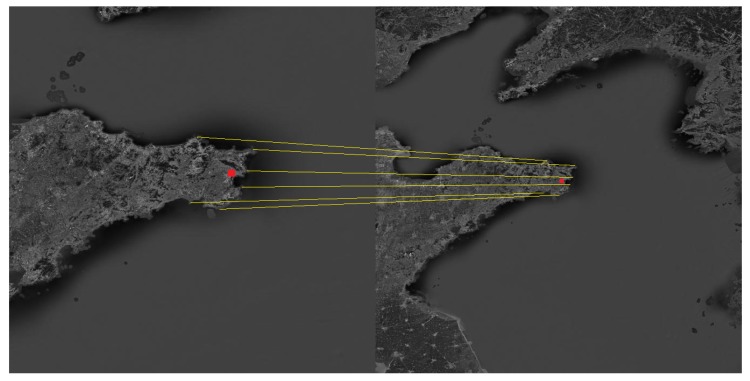
Schematic diagram of the matching process about the Shandong Peninsula.

**Figure 6 sensors-19-02917-f006:**
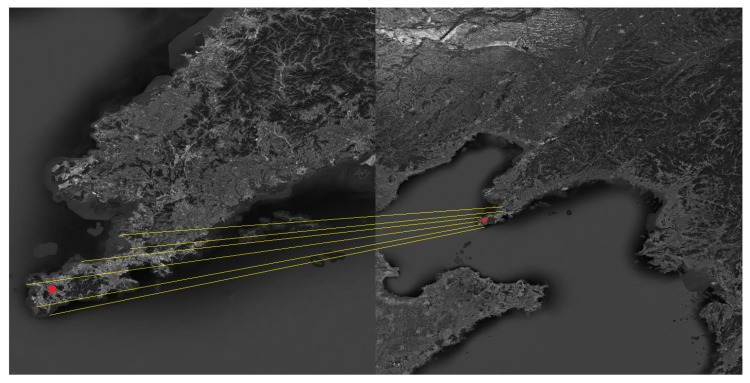
Schematic diagram of the matching process about the Liaodong Peninsula.

**Figure 7 sensors-19-02917-f007:**
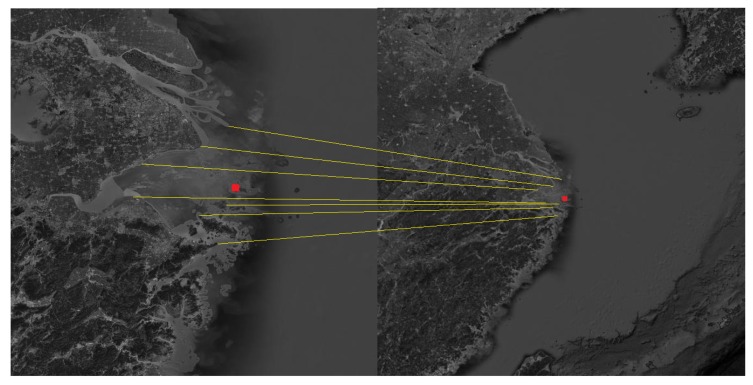
Schematic diagram of the matching process about the Yangtze River Delta.

**Figure 8 sensors-19-02917-f008:**
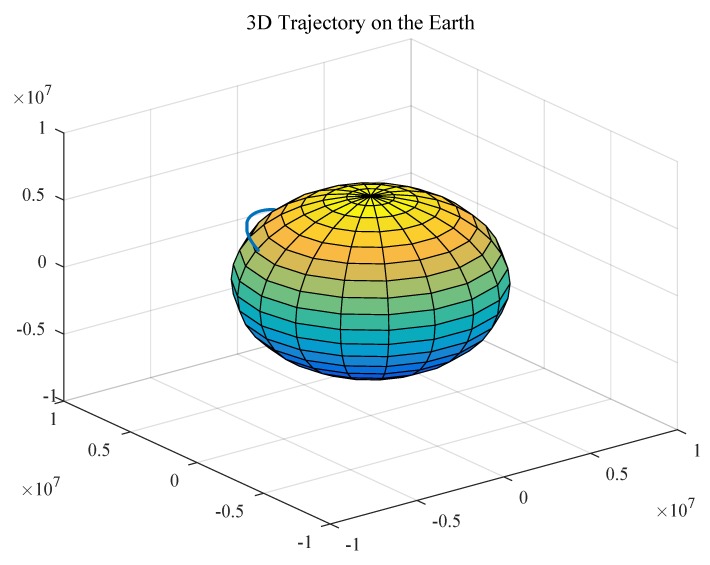
3D trajectory of the aircraft on the Earth.

**Figure 9 sensors-19-02917-f009:**
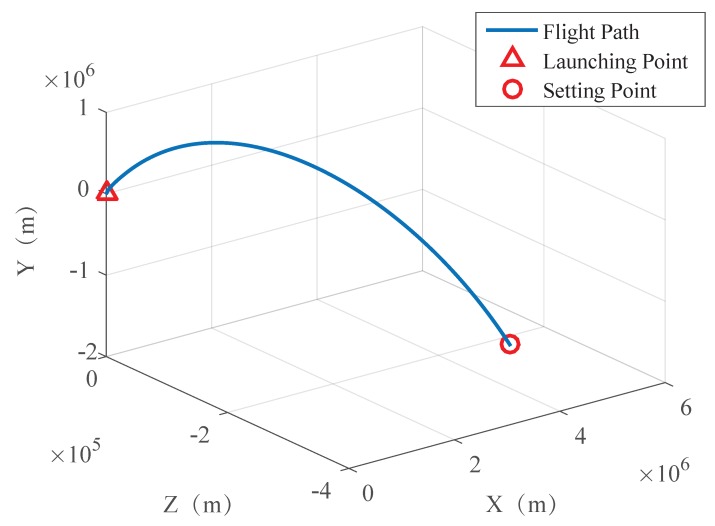
Partial 3D trajectory of the aircraft in the flight area.

**Figure 10 sensors-19-02917-f010:**
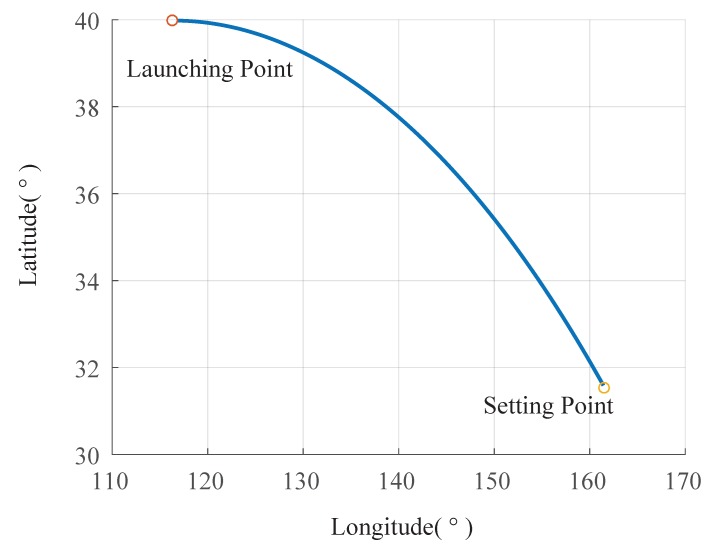
Trajectory of the aircraft under latitude and longitude reference.

**Figure 11 sensors-19-02917-f011:**
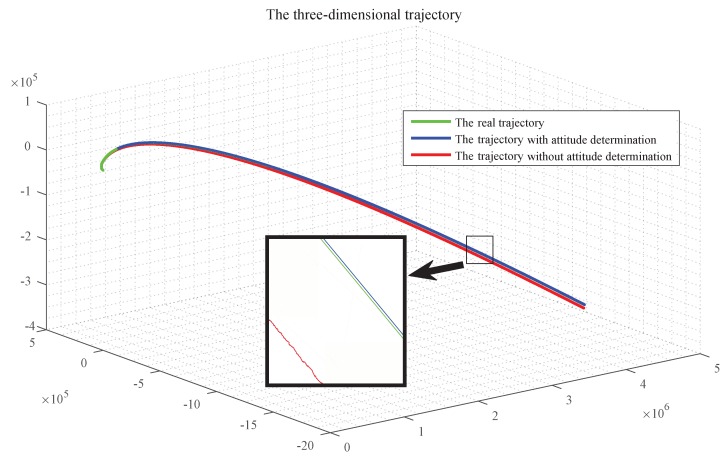
The Influence of the attitude determination on the accuracy of aircraft orbit.

**Figure 12 sensors-19-02917-f012:**
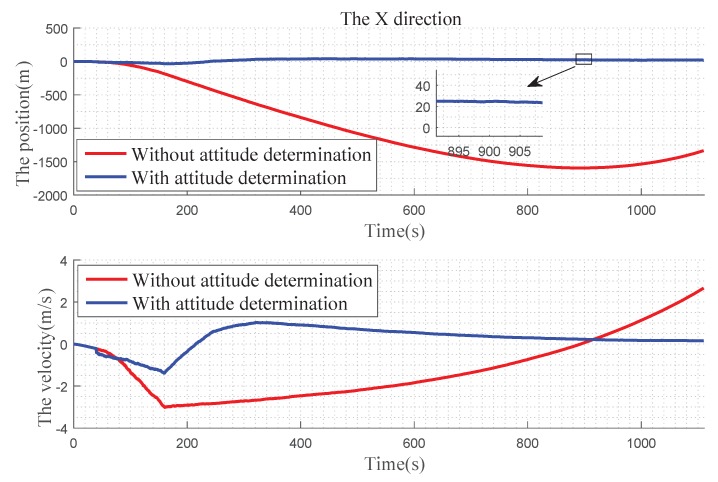
Precision comparison of the X direction.

**Figure 13 sensors-19-02917-f013:**
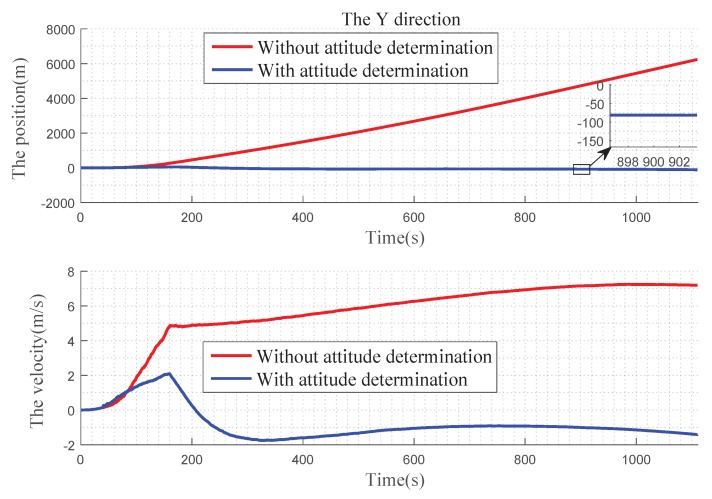
Precision comparison of the Y direction.

**Figure 14 sensors-19-02917-f014:**
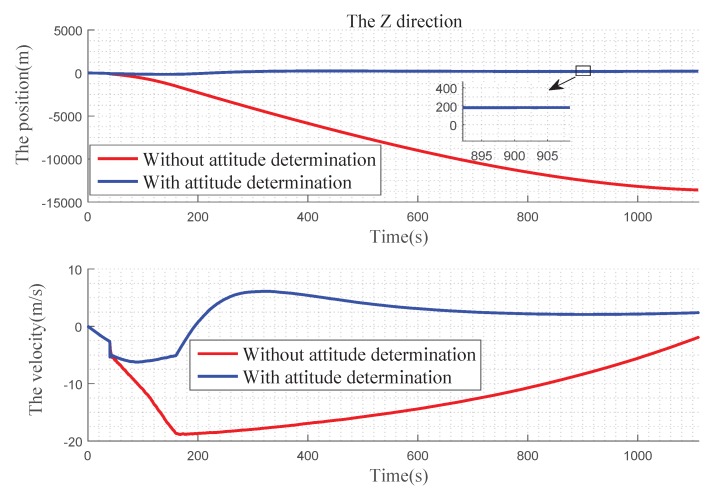
Precision comparison of the Z direction.

**Table 1 sensors-19-02917-t001:** The effect of the number and correlation of landmarks on the accuracy of the transformation matrix.

		3	4	5	6	7
Non-correlation	Pitch angle error	3.4146${}^{\prime \prime }$	2.0300${}^{\prime \prime }$	2.3630${}^{\prime \prime }$	1.6049${}^{\prime \prime }$	1.8437${}^{\prime \prime }$
	Yaw angle error	4.5328${}^{\prime \prime }$	1.4792${}^{\prime \prime }$	1.9401${}^{\prime \prime }$	1.8308${}^{\prime \prime }$	0.6509${}^{\prime \prime }$
	Roll angle error	1.5083${}^{\prime \prime }$	2.2644${}^{\prime \prime }$	0.8618${}^{\prime \prime }$	0.7915${}^{\prime \prime }$	0.2807″
Strong correlation	Pitch angle error	28.4667″	24.9020${}^{\prime \prime }$	13.8017${}^{\prime \prime }$	11.8040${}^{\prime \prime }$	3.3160${}^{\prime \prime }$
	Yaw angle error	7.9635${}^{\prime \prime }$	2.8863${}^{\prime \prime }$	4.4863${}^{\prime \prime }$	4.6482${}^{\prime \prime }$	4.0131${}^{\prime \prime }$
	Roll angle error	3.1885${}^{\prime \prime }$	5.7244${}^{\prime \prime }$	3.5935${}^{\prime \prime }$	2.2859${}^{\prime \prime }$	2.3123${}^{\prime \prime }$

**Table 2 sensors-19-02917-t002:** Position accuracy data statistics (unit: m).

		*X*	*Y*	*Z*
With attitude determination	The average error	25.6573	60.1919	175.0093
	The maximum error	40.5812	116.1984	240.7109
Without attitude determination	The average error	1004.2	2606.1	7664.0
	The maximum error	1601.2	6251.0	13615

**Table 3 sensors-19-02917-t003:** Velocity accuracy data statistics (unit: m/s).

		$V_{X}$	$V_{Y}$	$V_{Z}$
With attitude determination	The average error	0.5100	1.1333	3.4702
	The maximum error	1.2567	2.0557	6.2684
Without attitude determination	The average error	1.6633	5.5829	12.1824
	The maximum error	3.1103	7.2776	19.0498

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
