# Peer review of "SINS/Landmark Integrated Navigation Based on Landmark Attitude Determination"

_sensors, 2019, doi:10.3390/s19132917_

Round 1
Reviewer 1 Report
Dear authors,
thank you for the possibility to read the paper entitled: SINS/Landmark Integrated Navigation based on Landmark Attitude Determination.
I have some comments on the paper:
1) The abb. CNS in the abstract is not explained,
2) Fig. 1 should be explained better.
3) Page 10, the position error on three axes are denoted the same as the three random constant drift of gyroscope.
4) Line 250: the height is 0 m. Why 0 m? Is it because of mean sea level, or why?
5) Page 16, the gyroscope and accelerometer random noise drift satisfies a normal distribution with a mean of 0. Why these parameters? Why is the mean value 0? It does not demonstrate the real world. The parameters of simulation can not simulate the real conditions.
6) Fig. 12, 13, 14 should be better explained, in the scheme, the landmark is used only for position and attitude, but the velocity is compared. Also, in the description of the figure, the information, that it is error missing. I hope it is the error.
7) In References, some errors are present. More newer citations should be added.
The paper seems to be interesting, but it is based on simulated data. If the paper is based only on simulations, the results should be analyzed more and wide discussion should be done. The results of this paper are not good enough.
Table 1 should be analyzed better; the results are not clear.
Best regards,
Reviewer
Author Response
The authors thank the reviewer for the encouragement for this article. As for the improvements, we have made the following changes:
1) Since the manuscript explained all the abbreviations at the end, the meaning of the words such as CNS is not explained in the original text. But after analyzing the reviewer's opinion, I think that adding the explanation in the text will be more perfect, so I added a comment of the CNS in the abstract.
2) Considering that the acquisition of landmarks is related to optical technologies such as optical imaging and visualization, these are not the focus of this manuscript, so the previous manuscript wrote this part briefly. In the revision process, We will add the interpretation of the image to make the description of the landmark acquisition process more complete.
3) The three random constant drift of gyroscope has been wrongly written because of the careless mistake. Now we have corrected it.
4) Since the launch inertial system is used in the coordinate conversion process of the manuscript, in the simulation, the initial value here is selected as the launch inertial system, and the height of the launch inertial system coordinate origin can generally be set to 0.
5) In fact, the gyroscope and accelerometer errors are composed of two parts - constant bias and random drift. If they are treated as a whole error, the mean value is not 0; But when they are separated, they can be written like this.
6) In the simulation part, there are too few descriptions for Figs. 12, 13, and 14. We will add some more descriptions in detail in the revision. In addition, the ultimate goal of navigation is to determine the position and velocity of the aircrafts, the accuracy of navigation is also compared from position error and velocity error. And the principle of location and attitude determination is the way we navigate by building mathematical models, so there will be position and velocity comparisons but no attitude comparison in the simulation. Finally, the integrated navigation error is not 0, but only a small fluctuation in the vicinity of 0, I will improve the figures in the simulations.
7) In the references, many citations are incomplete, we will supplement them in the revision, and then quote some latest references.
8) For the results of Table 1, we will continue to supplement it, and continue to enrich theoretical analysis, making the results more intuitive and convincing.
Finally, I would like to thank the reviewer for your pertinent comments, and I will continue to improve this manuscript.
Reviewer 2 Report
1.The results on the quality for the number of landmark and using attitude information seems to be natural.
More extensive analysis on the proposed method should be required.
Comparison with other papers using the landmark should be included in the manuscript.
2. Too many typos and mis-spelling are found in the manuacript.
3. Reference format should follow the author's guide.
Author Response
The authors thank the reviewer for the encouragement for this article. As for the improvements, we have made the following changes:
1. In the process of determining the position by using the landmarks, in general, a three-dimensional coordinate has three unknowns elements, this makes the number of observable landmarks should be three or more. According to the reference, three landmarks that are not collinear can be used as the minimum basis for position determination, and have certain completeness; However, for the process of attitude determination, since the solution of the transformation matrix is calculated by the LS method using multiple samples, the relationship between the selection of the landmarks and the accuracy of the attitude needs to be determined by the equations of this paper. Therefore, this is not the same as the position determination. In this regard, this manuscript makes comparison with other papers and adds more theoretical analysis to the revision.
2. As for the typos and mis-spelling in this paper, I feel so sorry for this situation, it may be caused by careless writing and checking. I will check this manuscript word by word in the revision to find out all the mistakes and then correct them.
3. In the references, I find that some information of the quotes is incomplete. For this, I intend to supplement the information of the references according to the standard, and then quote some recent references.
Finally, I would like to thank the reviewer for your pertinent comments, and I will continue to improve this manuscript.
Round 2
Reviewer 1 Report
Dear authors,
Thank you for your answer and comments.
Best regards,
Reviewer
Reviewer 2 Report
The authors revised the manuscript well to the comments.